# Carbon Management for Intelligent Community with Combined Heat and Power Systems

Yongsheng Cao [1,2,3], Caiping Zhao [4,*] and Demin Li [1]

1    College of Information Science and Technology, Donghua University, Shanghai 201620, China;
     yongshengcao@mail.dhu.edu.cn (Y.C.); deminli@dhu.edu.cn (D.L.)
2    Department of Intelligent Science and Information Law, East China University of Political Science and Law,
     Shanghai 200042, China
3    China Institute for Smart Court, Shanghai Jiao Tong University, Shanghai 200030, China
4    Shanghai South of City Power Supply Company, Shanghai 201100, China
*    Correspondence: saberzcp@gmail.com

**Abstract:** In recent years, solar power technology and energy storage technology have advanced, leading to the increased use of solar power devices and energy storage systems in residential areas. Carbon management has become an important method to help the community manager guide energy consumption in a timely manner, effectively reduce the carbon emissions of the community, and reduce the substantial harm to the environment. This paper aims to study the issue of carbon management and resource allocation in an intelligent community with combined heat and power (CHP) systems and solar power. The presence of heterogeneous load demands in the power grid was considered. The main objective was to minimize the average system cost over time, which included the costs associated with the power grid and gas. The Lyapunov optimization theory was employed to solve the non-convex optimization problem of carbon management and resource allocation without energy sharing. To solve the energy-sharing problem, we designed an energy-sharing algorithm based on the Q-learning algorithm. Lastly, we conducted extensive simulations using actual trace data to validate the effectiveness of our proposed algorithms.

**Keywords:** resource allocation; carbon management; energy sharing; Lyapunov optimization; Q-learning

## 1. Introduction

As solar power technology advances, many solar power storage devices have emerged in the intelligent community. The intelligent community utilizes emerging technologies such as smartphones, embedded computers, network technology, and radio frequency identification technology to make the entire community management more intelligent and improve the quality of life of residents [1]. The total energy consumption of the construction industry in China accounted for 45.5% of the national energy consumption in 2020 according to China Building Energy Efficiency Annual Development Research Report 2022. Considering the growing energy demand and the necessity to reduce reliance on fossil fuels, solar power sources have gained increased attention. Combined heat and power (CHP) systems become increasingly popular in the intelligent community because of their ability to generate both electricity and thermal energy simultaneously, with relatively low carbon emissions. The CHP system mentioned in this paper is the internal combustion engine CHP system. Compared to previous methods that generate electricity and thermal energy separately, combining them can lead to a significant improvement in the efficiency and cost-effectiveness of energy generation. The utilization of photovoltaic-driven generators for electricity, and micro-combined heat and power (micro-CHP) for both electricity and heat, as well as energy storage system (ESS) and hot water storage tanks, can help reduce the energy demand of buildings [2]. Ensuring a low-carbon and reliable energy supply is a crucial responsibility for the development of smart cities.

There are two types of loads in the power grid: elastic load (EL) and rigid load (RL). Elastic loads are those whose demand can change with the electricity price. For example, air conditioners and electric water heaters are elastic loads. Rigid loads are those whose demand does not change with the electricity price. For example, lighting and industrial production are rigid loads. Elastic loads can be used to adjust the balance of the power grid by adjusting the electricity price, whereas rigid loads cannot. Therefore, the power grid needs to have a certain proportion of elastic loads to ensure its stable operation.

In the following two paragraphs, we briefly discuss the related work about the carbon management for intelligent community with CHP. The proximity of multiple energy vectors, such as electricity and heat, presents opportunities for the integration of energy systems and real-time management of multiple energy sources. There are some related works about the energy management of CHP systems to improve the efficiency of the power system. Song et al. [3] indicated that primary energy consumption can be significantly reduced, with the potential for even greater performance improvements through joint coordination. The operation strategy of the CHP system and the types of buildings have a big influence on the benefits of energy sharing. A novel approach to heat and power management has been devised in [4], enabling CHP systems to operate with greater flexibility, which allows CHP systems to independently optimize their power output while taking into account the degradation of the fuel cell, ultimately resulting in increased profitability for users. The method of enhancing the flexibility of CHP systems is investigated in [5] through the refinement of ramping and reserve modeling constraints. These related works did not consider the heating compensation mechanism between CHP units and heat storage. To achieve this, the paper suggests revised ramping constraints that take into account the internal mechanical structure of CHP units and proposes the regulation of heat exchange rates through the use of a heating butterfly valve. Additionally, a new model for available reserve capacity is introduced, which considers the heating compensation mechanism between CHP units and heat storage.

As information and communication technology continues to evolve, the communication between smart appliances and control centers has improved significantly, enabling the implementation of demand response strategies like real-time pricing to schedule tasks and ultimately reduce costs [6]. Smart appliances have the capability to arrange their tasks based on spot price strategies, which allows them to avoid the peak [7]. Some related works have designed algorithms to assist practical systems in making decisions according to energy storage, utilization, or selling to the grid based on real-time prices (spot prices). An energy management strategy for CHP was proposed in [8] with the demand response according to electricity price. A combination of power units and heat exchange stations in heating systems was proposed in [9] to improve the flexibility of system and reduce operating costs. An optimal scheduling method for a microgrid (MG) with CHP system using model predictive control is proposed in [10] to improve its efficiency and economic performance. The next four papers assumed predictability of future electricity prices and load demand, and focused solely on proposing optimal task scheduling algorithms according to consumer convenience. The paper [11] proposes a control strategy to schedule power devices of users in a smart microgrid, aiming to maintain smooth power balancing and ensure system stability under uncertain generation and load conditions. A solution is proposed in [12] to standardize and automate the tests for solar power-based generators. A three-layer collaborative optimization model is proposed in [13] to comprehensively the relationships among users, edge nodes, a cloud center, and a multi-edge league. The paper [14] proposed a regulating region method to accurately describe the heating-restricted reserve capacity of the Combined Heat and Power (CHP) units. Additionally, an integrated power and heat dispatch approach is developed, which utilizes the regulating region to formulate the available CHP reserve capacity. Some literature uses reinforcement learning methods to solve energy management problems. A total cost of ownership model was established in [15] including energy consumption and power source degradation, where the Q-learning algorithm is proposed to determine the optimal energy management strategy. A real-time

energy management strategy was proposed in [16] by combining the Q-Learning method with the model predictive control method. However, these papers did not consider carbon management with the heterogeneous load demands. This paper aims to study the issue of carbon management and resource allocation with CHP for heterogeneous load demands in an intelligent community to minimize the system costs associated with power grid and gas.

Three contributions of the paper are summarized as follows.

- Our study introduces an integrated model comprising a CHP system, a solar panel, an ESS, and a boiler. The primary aim of our research is to minimize the overall cost associated with grid usage, including expenses related to the power grid and natural gas (natgas) consumption. To achieve this objective, we propose a non-sharing resource allocation algorithm that utilizes the Lyapunov optimization method specifically designed for grids with EL and RL.
- Our proposed strategy focuses on cooperative energy sharing within the smart grid, aiming to implement an effective energy-sharing approach. This approach forms the foundation of our cooperative solar power-sharing algorithm, which utilizes the Q-learning algorithm. Under this algorithm, each MG is required to communicate with neighboring units through a centralized controller.
- Simulations have been conducted by using actual trace data to verify the effectiveness of our proposed algorithms. The energy-sharing algorithm was compared with a non-sharing resource allocation algorithm, and the results showed that our approach could decrease the economic cost by almost 19% while still fulfilling the energy demands of all residents.

The remainder of this article is divided into four sections. A mathematical model for a power grid that incorporates CHP systems, solar panels, ESS, and boilers was presented in Section 2. The solar power sharing strategy and control objectives was introduced. Section 3 formulates an optimization problem with certain constraints in both non-sharing and cooperative grid settings and explains the specifics of our algorithm design. In Section 4, we do the simulation of our non-sharing and sharing algorithm with the actual data. Finally, in Section 5, we draw conclusions based on our findings.

## 2. System Model and Problem Formulation

The system model of the intelligent community is shown in Figure 1, where there are CHP, boiler, solar equipment, ESS in MG $k = \{1, 2, \cdots, N_t\}$ in time slot $t$. We depict the power flow, information flow, and carbon flow in the different legends.

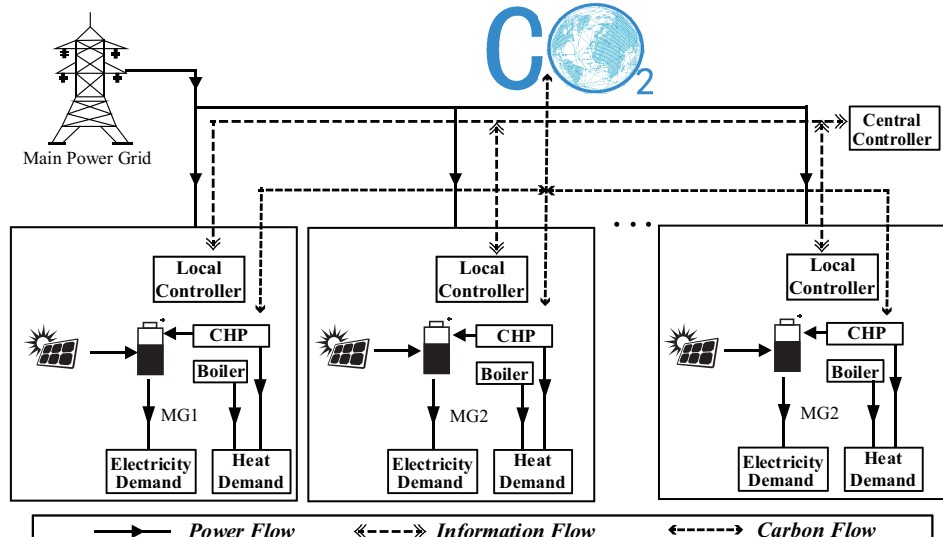

**Figure 1.** The power flow, information flow, and carbon flow in the intelligent community.

We posit that for a given time slot $t$, the CHP system consumes natgas denoted as $u^k(t)$. Concurrently, the CHP system within MG $k$ generates electric energy $\eta_e u^k(t)$ for the battery and thermal energy $\eta_h u^k(t)$ to meet the heating demand. The conversion efficiencies from natgas to electricity and heat are represented by $\eta_e$ and $\eta_h$, respectively. Additionally, the battery derives energy $r_s^k(t)$ from solar power, whereas the boiler dispatches energy $g^k(t)$ to fulfill the heating requirement. The electricity price, $\lambda_e(t)$, is constrained within the range $[\lambda_{e,\min}, \lambda_{e,\max}]$. We maintain the assumption that the natgas price $\lambda_g$ remains constant across all time slots due to its relative stability. Both the electricity price $\lambda_e(t)$ and the natgas price $\lambda_g$ are acquired from PG&E [17]. The algorithm aims to minimize the system cost by focusing on the energy $p^k(t)$ sourced from the power grid and the natgas $u^k(t)$ used by the CHP system along with $g^k(t)$ used by the boiler. In the following part, we formulate the mathematical model of solar power, electricity, and heat demand, ESS, the energy-sharing strategy, and the objective function.

## 2.1. Solar Power

Community solar refers to a solar energy system that is shared by multiple households or businesses within a community. Instead of installing solar panels on individual rooftops, a community solar project typically involves installing a larger solar array in a centralized location, such as a field or parking lot, and distributing the electricity generated to multiple subscribers. We set the solar energy from the renewable energy device as $r_s(t)$ for MG $k$,

$$r_s^k(t) \leq r_s^{k,\max}, \tag{1}$$

where $r_s^{k,\max}$ is the upper bound of the solar energy. The solar energy upper bound is the maximum amount of power that a solar energy system can generate at a given time and location. The solar energy upper bound depends on many factors, including the number, size, and type of solar panels, as well as the intensity and duration of sunlight.

## 2.2. Power and Heat Demand

For MG $k$, we establish the electric power demand as $e^k(t)$ and the thermal demand as $h^k(t)$. We operate under the assumption that tasks are continuous, with each task consuming electricity at a steady rate represented by $\pi_k^t$. The electricity demand for time slot $t$ is met by the joint contributions of the external grid, represented by $p^k(t)$, and the battery, designated by $b^k(t)$. On the other hand, the CHP system can produce heat $\eta_h u^k(t)$ to meet the heat demand, and the remaining heat demand $\eta_s g^k(t)$ is met by the boiler, with $\eta_s$ symbolizing the efficiency of heat generation from natgas consumption within the boiler. Then we can construct the following relationship,

$$e^k(t) = p^k(t) + b^k(t), \tag{2}$$
$$h^k(t) = \eta_h u^k(t) + \eta_s g^k(t). \tag{3}$$

## 2.3. Elastic Load and Rigid Load

Elastic loads have emerged with the introduction of smart appliances, which enable users to schedule loads at their convenience. For each load in microgrid (MG) $k$, we need to focus on the load's required time $a_k^t$ and the load's cutoff time $d_k^t$. It is crucial that the load is completed before the deadline $t + d_k^t$. If the required time $a_k^t$ equals the deadline $d_k^t$, the load needs to be fulfilled immediately, indicating its intolerance to delay. On the contrary, if the load demand can be met in a time frame shorter than the deadline, it is considered EL. The focus of our study is to discuss the optimal scheduling algorithm for EL. The delay is denoted as $s_k^t$, to represent the time of postponement. For RL, the delay $s_k^t$ is set to 0. Additionally, we define the parameter $d_{\max} = \max_{t,k} d_k^t$ as the maximum deadline among all loads and time slots.

### 2.4. Carbon Emission Constraint

Both electricity consumption from the grid and natural gas usage are associated with carbon emissions. The electricity generated on the grid comes primarily from sources such as coal, natural gas, nuclear, hydro, wind, and solar power. Among these sources, coal and natural gas are the primary fuels used for electricity generation, and their combustion releases greenhouse gases such as carbon dioxide, resulting in carbon emissions associated with grid electricity consumption. Natural gas is a relatively clean fossil fuel, and its combustion produces less carbon dioxide emissions than coal. However, it still contributes to carbon emissions. Additionally, the extraction, processing, transportation, and combustion of natural gas can also produce methane and other greenhouse gases, which can have an impact on climate change. To reduce carbon emissions, we can adopt measures such as energy conservation, using clean energy sources, and reducing natural gas usage to mitigate the environmental impact of electricity consumption and natural gas usage. The carbon dioxide emissions from electricity generation and that from natural gas are set as $\omega_p$ and $\omega_g$, then we have the following equation according to [18],

$$\omega_p p^k(t) + \omega_g g^k(t) \leq C_k^{Em}, \tag{4}$$

where $C_k^{Em}$ is the peak carbon dioxide emissions from MG $k$. The peak carbon dioxide emissions from MG refer to the highest point of carbon dioxide emissions produced by the MG during its operation. By monitoring and evaluating the peak carbon dioxide emissions of an MG, its energy utilization efficiency and environmental impact can be assessed, and its energy configuration and management strategies can be designed and optimized to achieve lower carbon emissions and higher energy utilization efficiency.

### 2.5. Energy Storage System

We do not take into account any electricity losses during the charging and discharging process. Figure 1 illustrates that the energy stored in the ESS during time slot $t$, denoted as $B^k(t)$ for MG $k$, comprises three components. The first component represents the energy obtained from the external grid, the second component represents the energy generated by solar power, and the last component represents the energy supplied by the CHP system. Consequently, the battery level $B^k(t)$ during time slot $t$ can be calculated using the following equation:

$$B^k(t+1) = B^k(t) + r_s^k(t) + \eta_e u^k(t) - b^k(t). \tag{5}$$

In practical applications, the power value $p^k(t)$ is generally positive. In this context, we observe that the battery level $b^k(t)$ is expected to be less than or equal to the energy stored in the ESS. The constraints can be described as follows:

$$|b^k(t)| \leq b_{\max}^k, \tag{6}$$

$$b^k(t) \leq \min\{B^k(t), e^k(t)\}, \tag{7}$$

where $b_{\max}^k$ represents the maximum charging rate in MG $k$. The constraints (6) and (7) indicate that the amount of energy charged or discharged from ESS has the range limit.

### 2.6. Energy-Sharing Strategy

Interconnection between different MGs enables energy sharing. We set the energy sharing between MG $k$ and MG $j$ in the time slot $t$ as $\varsigma_j^k(t)$. The power demand $e^k(t)$ can be met by the energy in MG $k$ from ESS and the neighbor MG in the following expression:

$$p^k(t) = e^k(t) - b^k(t) + \sum_{j \neq k} \varsigma_j^k(t). \tag{8}$$

MG $k$ can share the electricity $r_j^k(t)$ from other nearby MG $j$. Energy sharing between microgrids is an internal variable, and the total shared energy should sum up to zero. We have $\sum_k \sum_{j \neq k} \varsigma_j^k(t) = 0$.

*2.7. Objective Function*

In each time slot $t$, the overall cost of our system comprises two components: the electricity cost incurred from the power grid and the natgas cost by the CHP system and the boiler. From time slot $t$ to $t - d_{\max} + 1$, the electricity demand $e^k(t)$ and the heat demand $h^k(t)$ are satisfied. Our objective is to develop a scheduling algorithm that minimizes the long-term average cost by optimizing the allocation of electricity and natural gas. We have the following average cost in the long term,

$$f_{tol}^{avg} = \lim_{T \to \infty} \frac{1}{T} \sum_{t=1}^{T} \mathbb{E}\{\sum_{i=1}^{N} p^k(t)\lambda_e(t) + u^k(t)\lambda_g + \eta_s g^k(t)\lambda_g\}. \tag{9}$$

For the sake of simplicity, we have intentionally omitted certain practical factors, such as electricity loss during transmission. The purpose of this omission is to concentrate on minimizing electricity sourced from the grid, focusing on variables such as ESS charging rate $b(t)$, and the energy generated by CHP $u(t)$. We study a non-sharing scenario where energy is not shared with nearby MG under carbon emission constraints. Given the current state of the system, we aim to design an optimal control strategy that is not complicated and difficult to calculate. Elastic loads lead to dissatisfaction and the dissatisfaction will be larger when the elastic loads delay more time. We denote the ELs' dissatisfaction function $F_k^t(s)$ for the delay $s$ in MG $k$. The relationship between the dissatisfaction function of elastic loads and the time delay is that as the time delay increases, the dissatisfaction function of elastic loads also increases. This means that the more delayed elastic loads are in receiving service, the more dissatisfied they are. The ELs' dissatisfaction function is limited by $\alpha$.

$$\lim_{T \to \infty} \sup \frac{1}{T} \sum_{t=1}^{T} \sum_{k=1}^{n_t} F_k^t(s_k^t) \leq \alpha, \tag{10}$$

where $s_k^t$ is the delay for EL in MG $k$. Tasks must be completed before their deadlines, and they are only scheduled if there is enough time, and the delay $s_k^t$ has the following constraint,

$$0 \leq s_k^t \leq d_k^t - a_k^t. \tag{11}$$

We can simplify this optimization problem under carbon emission constraints as P1 by using Equations (2) and (3):

$$\text{P1}: \min_{\substack{r_s^k(t), s_k^t, \\ b^k(t), u^k(t)}} \lim_{T \to \infty} \frac{1}{T} \sum_{t=1}^{T} \mathbb{E}[(e^k(t) - b^k(t))\lambda_e(t) + u^k(t)\lambda_g$$

$$+ (h^k(t) - \eta_h u^k(t))\lambda_g]$$

$$\text{s.t.} \quad \omega_p(e^k(t) - b^k(t)) + \omega_g g^k(t) \leq C_k^{Em}, \tag{12}$$

$$B^k(t+1) = B^k(t) + r_s^k(t) + \eta_e u^k(t) - b^k(t), \tag{13}$$

$$|b^k(t)| \leq b_{\max}^k, \tag{14}$$

$$b^k(t) \leq \min\{B^k(t), e^k(t)\}, \tag{15}$$

We denote that $\lim_{T\to\infty}\sum_{t=1}^{T}e^k(t)\lambda_e(t)$ is the total operational cost of electricity demand. Then we have the following equation under carbon emission constraints:

$$\lim_{T\to\infty}\sum_{t=1}^{T}e^k(t)\lambda_e(t) = \lim_{T\to\infty}\sum_{t=1}^{T}\sum_{i=1}^{n_t}\sum_{j=0}^{a_k^t-1}\pi_k^t\lambda_e(j+t+s_k^t), \qquad (16)$$

where $\pi_k^t$ is the electricity consumption rate. We rewrite the optimization problem P1 as the following problem P2:

$$\text{P2}: \min_{\substack{r_s^k(t),s_k^t,\\b^k(t),u^k(t)}} \lim_{T\to\infty}\frac{1}{T}\sum_{t=1}^{T}\mathbb{E}[\sum_{i=1}^{n_t}\sum_{j=0}^{a_k^t-1}\pi_k^t\lambda_e(j+t+s_k^t)$$

$$- b^k(t)\lambda_e(t) + \eta_e u^k(t)\lambda_g + h^k(t)\lambda_g]$$

$$\text{s.t.}\quad \omega_p(e^k(t)-b^k(t)) + \frac{\omega_g}{\eta_s}(h^k(t)-\eta_h u^k(t)) \le C_k^{Em}, \qquad (17)$$

$$B^k(t+1) = B^k(t) + r_s^k(t) + \eta_e u^k(t) - b^k(t), \qquad (18)$$

$$|b^k(t)| \le b_{\max}^k, \qquad (19)$$

$$b^k(t) \le \min\{B^k(t), e^k(t)\}, \qquad (20)$$

We need to solve the optimization problem P2 under the constraint of the stability of the battery level and carbon emission constraints. We utilize the Lyapunov optimization method. Lyapunov optimization is a technique for finding the optimal control input for a system. It is based on the idea of finding a control input that minimizes the value of a Lyapunov function, which is a function of the state of the system. The Lyapunov function is used to measure the stability of the system, and the optimal control input is the one that minimizes the value of the Lyapunov function.

Lyapunov optimization requires the creation of virtual queues. A virtual queue is an abstract concept that can be used to describe any type of queue in a system. Virtual queues can be used to represent physical queues or they can be used to represent logical queues. Virtual queues can help us better understand the behavior of a system. By studying virtual queues, we can better understand the performance bottlenecks of a system, and we can find ways to improve those bottlenecks. By using virtual queues, we can avoid many of the problems that can occur in physical queues, such as queue congestion and deadlocks. We denote the following virtual queue $U(t)$,

$$U(t+1) = \max\{U(t) + \sum_{i=1}^{n_t}F_k^t(s_k^t) - \alpha, 0\}. \qquad (21)$$

We can prove that if this virtual queue $U(t)$ meets the restriction $\lim\sup_{T\to\infty}U(T)/T = 0$, then we have

$$\lim_{T\to\infty}\sup\frac{1}{T}\sum_{t=1}^{T}\sum_{i=1}^{n_t}F_k^t(s_k^t) \le \alpha. \qquad (22)$$

### 2.8. Resource Allocation Algorithm

In this part, we design the resource allocation algorithm under carbon emission constraints based on the Lyapunov optimization method. We can stabilize the battery level queue $B^k(t)$ and the virtual queue $U^k(t)$ by the Lyapunov drift. We design a function $L^k(t) = \frac{1}{2}[U^k(t)^2 + (B^k(t)-\theta)^2]$. We aim to minimize the drift of the function $L^k(t)$, which leads the stability of the battery level $B^k(t)$ closer to the constant $\theta$. We have denoted the parameters $n_{\max} = \max_t n_t$, $F_{\max} = \max_{t,i}F_t^i(d_t^i)$. By the Lyapunov approach, we set the variable $Z^k(t) = (U^k(t), B^k(t))$ and the Lyapunov drift $\Delta = E\{(L^k(t+1)-L^k(t))|Z^k(t)\}$.

**Lemma 1.** *The Lyapunov drift $\Delta$ will have the following property,*

$$
\Delta \leq U^k(t)\mathbb{E}[\sum_{i=1}^{n_t} F_k^t(s_k^t) - \alpha|Z^k(t)] + \frac{1}{2}[r_s^k(t) + \eta_e u^k(t) \tag{23}
$$
$$
- b^k(t)]^2 + (B^k(t) - \theta)(r_s^k(t) + \eta_e u^k(t) - b^k(t)) + \frac{1}{2}(n_{\max}^2 F_{\max}^2 + \alpha^2).
$$

*We aim to minimize the Lyapunov drift $\Delta$ to keep the virtual queue $U^k(t)$ and the battery level queue $B^k(t)$ stable. We denote the trade-off between the electricity and gas cost and the Lyapunov drift $\Delta$ as V. We add $f = V\mathbb{E}[\sum_{k=1}^{n_t}\sum_{j=0}^{a_k^t-1}\pi_k^t\lambda_e(j+t+s_k^t) - b^k(t)\lambda_e(t) + \eta_e u^k(t)\lambda_g + h^k(t)\lambda_g]$ on both sides of Equation (23). We design non-sharing resource allocation algorithm (NRA) by minimizing the right side of $\Delta + f$. During time slot t, we can calculate the delay $s_k^{t*} = \arg\min_{0 \leq s_k^t \leq d_k^t - a_k^t} U(t)F_k^t(s_k^t) + V\sum_{j=0}^{a_k^t-1}\pi_k^t\lambda_e(j+t+s_k^t)$. Let $f(t) = (B^k(t) - \theta)(r_s^k(t) + \eta_e u^k(t) - b^k(t)) + \frac{1}{2}(r_s^k(t) + \eta_e u^k(t) - b^k(t))^2 + V(\eta_e u^k(t)\lambda_g + h^k(t)\lambda_g - b^k(t)\lambda_e(t))$. To minimize the objective function $\mathbb{E}[f(x)|Z(t)]$, we have the following equation,*

$$
\mathscr{L} = \sum_{t=1}^{T}[(B^k(t) - \theta)(r_s^k(t) + \eta_e u^k(t) - b^k(t)) + \frac{1}{2}\iota_t(r_s^k(t)
$$
$$
+ \eta_e u^k(t) - b^k(t))^2] + \sum_{t=1}^{T}\zeta_t(\omega_p e^k(t) - \omega_p b^k(t) + \frac{\omega_g}{\eta_s}h^k(t) - \frac{\eta_h\omega_g}{\eta_s}u^k(t) - C_k^{Em})
$$
$$
+ \sum_{t=1}^{T}\mu_t(b^k(t) - b_{\max}^k),
$$

*where $\iota_t, \zeta_t, \mu_t$ are the Lagrange multipliers and dual variables for constraints (24b), (24c). This optimization problem is convex, feasible, and satisfies Slater's condition [19]. We solve the optimization problem by a standard primal–dual gradient method when $\zeta_t \geq 0$ or $\zeta_t = 0, \forall t$; we have,*

$$
r^*(t) = \delta_r(\frac{\partial L}{\partial r(t)}) = \delta_r(r(t) + B(t) - \theta - \iota_t + \zeta_t), \tag{24a}
$$

$$
u^*(t) = \frac{1}{\eta_e}\delta_u(\frac{\partial L}{\partial \eta_e u(t)}) = \frac{1}{\eta_e}\delta_u(\eta_e u(t) + B(t) - \theta - \iota_t + VC_g) + \frac{\zeta_t\eta_h\omega_g}{\eta_s}, \tag{24b}
$$

$$
b^*(t) = \delta_b(b(t) - (B(t) - \theta) - VC_e(t) + \iota_t + \mu_t - \omega_p\zeta_t), \tag{24c}
$$

$$
\zeta_t^* = \delta_\zeta(\omega_p e^k(t) - \omega_p b^k(t) + \frac{\omega_g}{\eta_s}h^k(t) - \frac{\eta_h\omega_g}{\eta_s}u^k(t) - C_k^{Em})_{\zeta_t}^+, \tag{24d}
$$

$$
\mu_t^* = \delta_\mu(b(t) - b_{\max})_{\mu_t}^+. \tag{24e}
$$

*where $\delta_r, \delta_u, \delta_b, \delta_\zeta, \delta_\mu$ are positive parameters. We denote the maximum electricity price as $\lambda_{e,\max}$ and the parameter $\theta = b_{\max} + V\lambda_{e,\max}$. We will always have $\theta - B(t) - \lambda_e(t) > 0$ when the battery level $B(t) < b_{\max}$. ESS will draw the energy from power grid and $b(t) = -b_{\max}$. ESS discharges when the battery level $B(t) > b_{\max}$. The battery level $B(t)$ and the system cost have the upper bound. We prove the performance of NRA in Theorem 1.*

**Theorem 1.** *We denote the parameter $\theta = b_{\max} + V\lambda_{e,\max}$ and the initial battery level $B(0) = \theta$, then the battery level $B(t)$ will have the following property:*

$$
0 \leq B^k(t) \leq \theta + b_{\max}^k + r_{\max}^k. \tag{25}
$$

**Proof.** First, we prove that the battery level $B(t)$ has the upper bound by using a mathematical induction method. We set the initial situation: in the time slot $t = 0$, we have the initial battery level $B(0) = \theta < \theta + b_{\max} + r_{\max}$. Then, we assume that the battery level $B(t) \leq \theta + b_{\max} + r_{\max}$. We continue to prove that $B(t+1) \leq \theta + b_{\max} + r_{\max}$. In the next time slot $t + 1$, we have the following two cases:

(1) If the battery level $B(t) \leq \theta$, ESS will discharge the maximum electricity when $b(t) = -b_{\max}$. Then we will obtain the battery level in the next time slot $B(t+1) \leq \theta + b_{\max} + r_{\max}$.

(2) If the battery level $B(t) > \theta$, we see that the charging rate $b(t) > 0$ from Equation (24) and ESS discharges. Then, we will have $B(t+1) \leq B(t) \leq \theta + r_{\max} + b_{\max}$. Above all, we prove that the battery level $B(t+1) \leq \theta + r_{\max} + b_{\max}$.

Second, we prove that the battery level $B(t)$ has a lower bound by mathematical induction. We assume that the battery level $B(t) \geq 0$ and we need to prove that $B(t+1) \geq 0$. In the next time slot $t + 1$, we have the following cases:

(1) If the battery $B(t) \leq \theta$, we see that the charging rate $b(t) > 0$ from Equation (24) and ESS charges. Therefore, we have the battery level $B(t+1) > B(t) \geq 0$.

(2) If the battery level $B(t) > \theta$, we have the property $B(t) > b_{\max} + V\lambda_{e,\max}$. The battery level $B(t+1)$ has the physical constraint $B(t+1) \geq 0$. Above all, we can prove that the battery level $B(t)$ has the lower bound and $B(t+1) \geq 0$. $\square$

**Theorem 2.** *The electricity and gas cost with NRA will satisfy the following property,*

$$
\begin{aligned}
\lim_{T \to \infty} \sup \frac{1}{T} \sum_{t=1}^{T} E[&\sum_{i=1}^{n_t} \sum_{j=0}^{a_k^t - 1} \pi_k^t \lambda_e(j + t + s_k^*) - b^*(t)\lambda_e(t) \\
&+ \eta_e u^*(t)\lambda_g + h(t)\lambda_g] \\
&\leq C + \lambda_{e,\max} b_{\max} + \frac{D + (b_{\max} + r_{\max})^2}{V}.
\end{aligned}
\tag{26}
$$

**Proof.** The constraint of charging amount $b(t)$ in the time slot $t$ is limited by $[0, b_{\max}]$ when ESS discharges. From Theorem 1, the battery level has the property $B(t) < \theta + b_{\max} + r_{\max}$, then we can obtain the property that $|\theta - B(t) - VC_e(t)| \leq b_{\max} + r_{\max} + VC_{e,\max}$.

$$
\begin{aligned}
\sum_{i=1}^{n_t} E[&U(t)F_k^t(s_k^t) + V \sum_{j=0}^{a_k^t - 1} \pi_k^t P(j + s_k^t + t)|Z(t) - \alpha U(t) \\
&+ E[(B(t) - \theta)(r(t) + \eta_e u(t) - b(t)) + \frac{1}{2}(r(t) + \eta_e u(t) \\
&- b(t))^2 + V(\eta_e u(t)\lambda_g + h(t)\lambda_g - b(t)\lambda_e(t))|Z(t)] \\
&\leq V\tilde{C} + (b_{\max} + r_{\max} + VC_{e,\max})b_{\max}.
\end{aligned}
\tag{27}
$$

We set $\tilde{C}$ as the lower bound of $C_e(t)$, and we can obtain the average value from time slot $t = 0$ to $T$ in the following equation.

$$
\begin{aligned}
E\{(L(t+1) - L(t))|Z(t)\} + VE[&\sum_{i=1}^{n_t} \sum_{j=0}^{a_k^t - 1} \pi_k^t \lambda_e(j + t + s_k^t) \\
&- b(t)\lambda_e(t) + \eta_e u(t)\lambda_g + h(t)\lambda_g] \\
&\leq TD + T(b_{\max} + r_{\max})^2 + VTC + VTC_{e,\max}b_{\max}.
\end{aligned}
\tag{28}
$$

By setting the initial battery level $B(0) = \theta$, we can figure out that $L(0) = 0$. We divide by $VT$ on both sides of Equation (28) and we have,

$$
\begin{aligned}
\lim_{T \to \infty} \frac{1}{T} \sum_{t=1}^{T} E[&\sum_{i=1}^{n_t} \sum_{j=0}^{a_k^t - 1} \pi_k^t \lambda_e(j + t + s_k^*) - b^*(t)\lambda_e(t) \\
&+ \eta_e u^*(t)\lambda_g + h(t)\lambda_g] \\
&\leq C + \lambda_{e,\max} b_{\max} + [D + (b_{\max} + r_{\max})^2]/V.
\end{aligned}
\tag{29}
$$

$\square$

From Equation (29), we can see that the total energy cost will show a converging trend with the increase in the parameter $V$. The Lyapunov optimization method ensures the stability and convergence of the system by designing the Lyapunov function. In the stable region, the value of the Lyapunov function is positive; in the unstable region, the value of the Lyapunov function is negative. By designing the Lyapunov function, the system can be guaranteed to be in a stable state in the stable region.

## 3. Energy-Sharing Algorithm

We denote the optimal total cost of the non-sharing resource allocation algorithm as $\lambda_{NRA}$ and the cost $\lambda_{NRA} = \sum_{i=1}^{N} \sum_{t=1}^{T} C_{tol}^{k}(t)$. We propose the energy-sharing algorithm (ESA) based on the Q-learning method. The total cost of the energy-sharing algorithm is denoted as $\lambda_{ESA}$. $\lambda_{ESA} = \sum_{i=1}^{N} \sum_{t=1}^{T} \hat{C}_{tol}^{k}(t)$, where $\hat{C}_{tol}^{k}(t)$ is denoted as the electricity and gas cost for MG $k$ in an energy-sharing scenario. We apply the Q-learning method to handle the resource allocation problem with energy sharing. The basic elements of Q-learning include: state space, action space, state transition probability, reward function, and value function. For the ESA, we define the following four elements:

(1) State Space

State space refers to the environment state where the agent is located, which can be discrete or continuous. We discretize the electricity price into $M$ intervals. The state space is set as $\Phi$ including electricity price and the number of MGs $N$, which is denoted as $\Phi = \{1, ..., M\} \times \{1, ..., N\}$.

(2) Action Space

Action space refers to the actions that the agent can take, which can be discrete or continuous. The maximum electricity shared with neighbor MGs is denoted as $E_{\max}$. The action space of the energy has three choices: drawn, hold on, and sharing:

$$A = \{-E_{\max}, 0, E_{\max}\} \tag{30}$$

We can obtain the equation $C_{NRA}^{k} - C_{ESA}^{k} = (\lambda_{NRA} - \lambda_{ESA}/N)$ by using the Nash bargaining method, according to Equation (32) in [20]. We define the temporary variable $tmp^{k} = C_{NRA}^{k} - \frac{\lambda_{NRA} - \lambda_{ESA}}{N}$. From Algorithm 2 in [20], MG $k$ draws energy from neighbor MGs when $\Delta tmp^{k} = tmp^{k} - \sum_{t=1}^{T} C_{tol}^{k}(t) \leq 0$ and shares energy with neighbor MGs when $\Delta tmp^{k} > 0$. The agent can choose one action at each time step, and draw or share energy from the neighbor MGs. The agent's goal is to minimize its total energy cost while satisfying the energy constraints.

(3) Reward Function

The reward function refers to the reward that the agent will receive after taking a given action in a given state. The reward function represents the reward received after taking an action in a given state. The value function represents the expected total reward received after taking an action in a given state. By continuously exploring the environment and updating the value function based on experience, the Q-learning algorithm can find the optimal policy. After executing an energy-sharing action with the state, a reward will come up. We can obtain the energy shared from neighbor MGs when the electricity price is high. We denote the reward function $rf_{t}(\phi, a)$ in the following equation,

$$rf_{t}(\phi, a) = \lambda_{e}(t) E_{\max} \frac{tmp^{k} - \sum_{t=1}^{T} C_{tol}^{k}(t)}{|tmp^{k} - \sum_{t=1}^{T} C_{tol}^{k}(t)|} \tag{31}$$

(4) Q-learning Algorithm

The Q-learning algorithm finds the optimal policy by continuously exploring the environment and updating the value function based on experience. We utilize Q-learning

to propose an update policy based on the action space, state space, reward, and value function. For each action–state pair $(\phi_k, a_k)$, we have the following value function:

$$Q_{t+1}[\phi_k, a_k] = Q_t[\phi_k, a_k] + \beta(rf_t(\phi, a_k) + \gamma Q_t[\phi'_k, a'_k] - Q_t[\phi_k, a_k]), \tag{32}$$

where state–action pair $(\phi'_k, a'_k)$ is the possible situation in MG $k$ in next time slot $t + 1$. There are two important parameters $\beta$ and $\gamma$. $\beta$ is the learning rate and $\gamma$ is the discount factor. The learning rate in Q-learning is a hyperparameter that controls how quickly the Q-learning algorithm updates the Q-table. A higher learning rate means that the Q-table will be updated more quickly, but it is also more likely to overfit. A lower learning rate means that the Q-table will be updated more slowly, but it is also less likely to overfit. The Q-learning algorithm typically uses a fixed learning rate. However, in some cases, using a dynamic learning rate may be more effective. A dynamic learning rate is a learning rate that changes over time. Dynamic learning rates can help the Q-learning algorithm avoid overfitting and learn more effectively in different environments. The discount factor in Q-learning is a hyperparameter that controls how much the Q-learning algorithm considers future rewards. A higher discount factor means that the algorithm will be more concerned with future rewards. This can help the algorithm avoid being too short-sighted and find more long-term strategies. The discount factor is usually set to a value between 0 and 1. A value of 0 means that the algorithm will only consider the current reward, whereas a value of 1 means that the algorithm will consider all future rewards. In practice, a discount factor of 0.9 or higher is often used. The discount factor is an important hyperparameter that can have a significant impact on the performance of the Q-learning algorithm. If the discount factor is set too low, the algorithm may be too short-sighted and unable to find effective strategies. If the discount factor is set too high, the algorithm may be too concerned with future rewards and unable to find balanced strategies. We summarize the energy-sharing algorithm (ESA) based on Q-learning in Algorithm 1 and the flow chart of energy-sharing algorithm in Figure 2.

---

**Algorithm 1** Energy-Sharing Algorithm (ESA)

---

1: **Initialization:** Initialize the Q-table with random values:
2: for state $\phi_k$ in all states $\Phi$:
3:    for action $a_k$ in all actions $A$:
4:       $Q[\phi_k, a_k]$ = random value
5: Initialize the learning rate $\beta = 0.6$ and the discount factor $\gamma = 0.85$.
6: **While True:**
7: Choose the energy-sharing action $a_k$ from the state $\phi_k$ by using $\theta$-greedy policy according to value function $Q$;
8: **Repeat** (for each step of episode):
9:    Obtain the close-optimal energy action $a_k$, the reward function $rf_t$, the state $\phi'_k$;
10:    Observe $a'_k$ from $\phi'_k$ by using $\theta$-greedy policy from $Q$;
11:    Update the value function $Q(\phi_k, a_k)$;
12:    Update the state $\phi_k \leftarrow \phi'_k$ and the action $a_k \leftarrow a'_k$ ;
13: **until** $\phi_k$ is terminal.
14: **end**

---

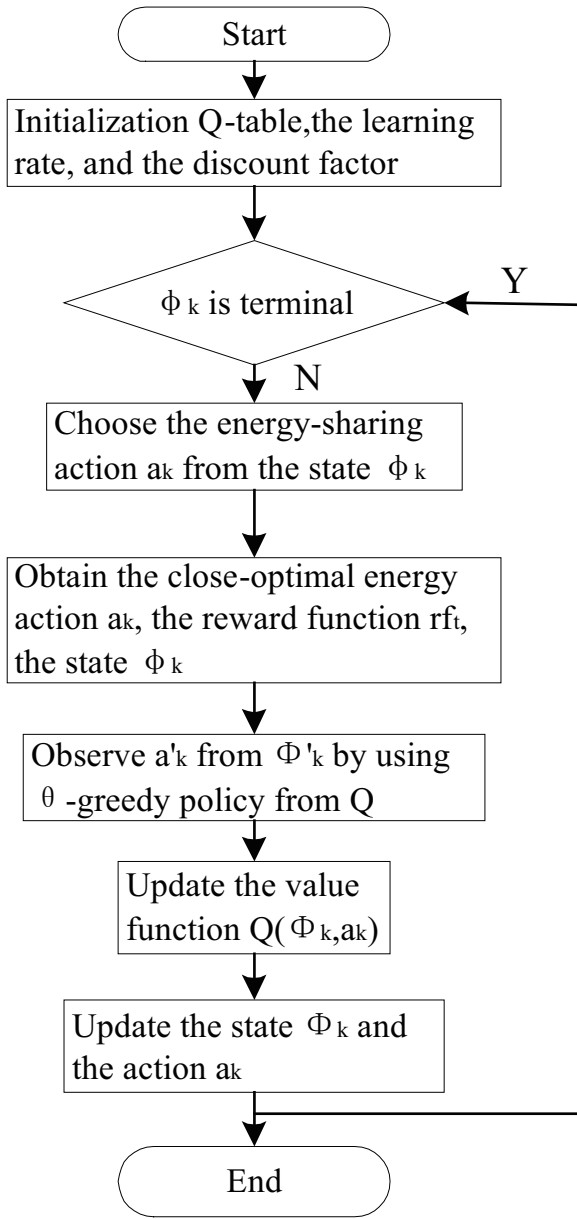

**Figure 2.** Flow chart of energy-sharing algorithm.

## 4. Numerical Simulation

In this section, we will evaluate the performance of the NRA algorithm and the ESA algorithm with real-time electricity price by Matlab 2021 on Intel Core i7. We list the parameters about the price, the capacity of ESS, the electricity, and heat demand in the following part.

In order to simulate and evaluate our algorithm, we need the following data: electricity prices, natural gas prices, electricity demand, and heat demand. Electricity price and natgas price are collected from Pacific Gas and Electric Company from 24 June 2023 to 28 June 2023. We show the electricity price in Figure 3 and we set each time slot to one hour. We calculate our total electricity consumption cost of RL loads and EL loads and draw some conclusions. We study a community with MGs with 200 appliances of 200 h and we set each time slot as 1 h. The overall efficiency of a CHP system in producing electricity is set as 80% and the maximum output of the CHP system is set as $u_{\max}^k = 8$ kWh. The capacity of ESS $b_{\max}$ is set as 6 kWh.

The carbon dioxide emissions from electricity generation and that from natural gas are denoted as $\omega_p = 450$ g $CO_2$/kWh and $\omega_g = 1.885$ kg $CO_2$/m$^3$. We have shown the parameter setting in Table 1.

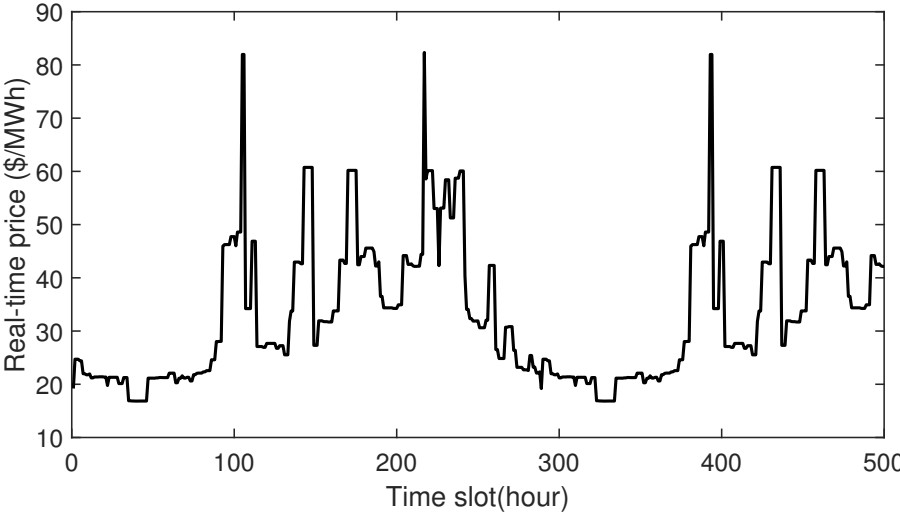

**Figure 3.** Real-time electricity price.

**Table 1.** Parameter setting.

| Parameter | Value |
|---|---|
| maximum output of CHP system $u_{max}^k$ | 8 kWh |
| capacity of ESS $b_{max}$ | 6 kWh |
| carbon dioxide emissions from electricity generation $\omega_p$ | 450 g $CO_2$/kWh |
| carbon dioxide emissions from natural gas $\omega_g$ | 1.885 kg $CO_2$/m$^3$ |
| price of natgas $\lambda_g$ | $USD5.4$/MMBtu |

The dissatisfaction function is a quadratic function of time, meaning that as time goes on, dissatisfaction increases. This may be because people become more impatient as they wait, or because they have higher expectations about the outcome of the wait. We denote the dissatisfaction function as a quadratic function $F(x) = x^2$. We set the price of natgas $\lambda_g$ as $USD5.4$/MMBtu.

After simulating the NRA, we have shown the electricity charged and discharged from ESS in Figure 4. ESS charges the electricity in the range $[0, b_{max}]$ and discharges the electricity at the discharging rate $b_{max}$. The battery level $B(t)$ in 200 time slots is shown in Figure 5 which has a hard constraint. By setting the parameter $\theta = b_{max} + VC_{e,max}$, the battery level of ESS $B(t)$ has an upper bound $\theta + b_{max} + r_{max}$. From Figure 5, we can see that the battery level of ESS $B(t)$ has a limit less than $\theta + b_{max} + r_{max}$ in 200 time slots. Under the Lyapunov optimization algorithm, the battery queue and dissatisfaction virtual queue are both stable. This means that at any given time, the number of batteries in the battery queue will not exceed the battery capacity, and the dissatisfaction in the dissatisfaction virtual queue will not exceed the maximum dissatisfaction. This stability is essential for ensuring the effectiveness of the battery queue and dissatisfaction virtual queue. If the number of batteries in the battery queue exceeds the battery capacity, then batteries may be depleted, leading to system failure. If the dissatisfaction in the dissatisfaction virtual queue exceeds the maximum dissatisfaction, then users may become dissatisfied with the system, leading to them leaving the system. The Lyapunov optimization algorithm ensures the stability of the battery queue and dissatisfaction virtual queue by using a Lyapunov function. A Lyapunov function is a function that measures the state of the system. If the value of

the Lyapunov function decreases over time, then the system is stable. The Lyapunov optimization algorithm uses the Lyapunov function to calculate the control inputs, which are used to make the value of the Lyapunov function decrease. By using the Lyapunov optimization algorithm, it is possible to ensure that the battery queue and dissatisfaction virtual queue are stable at any given time. This allows the system to effectively manage batteries and user dissatisfaction.

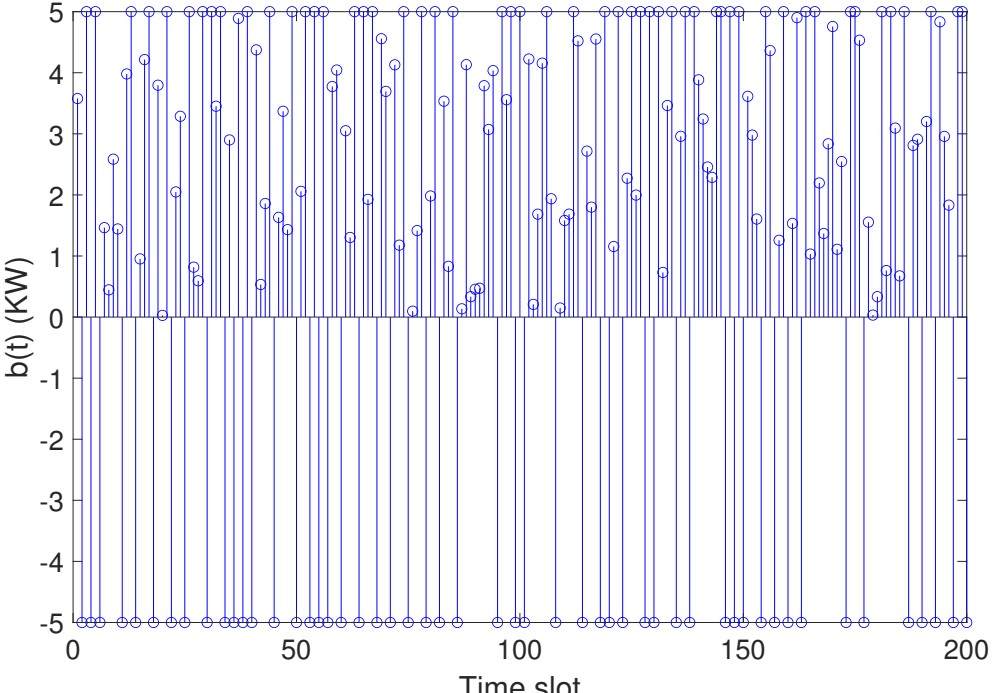

**Figure 4.** The result of electricity change in ESS.

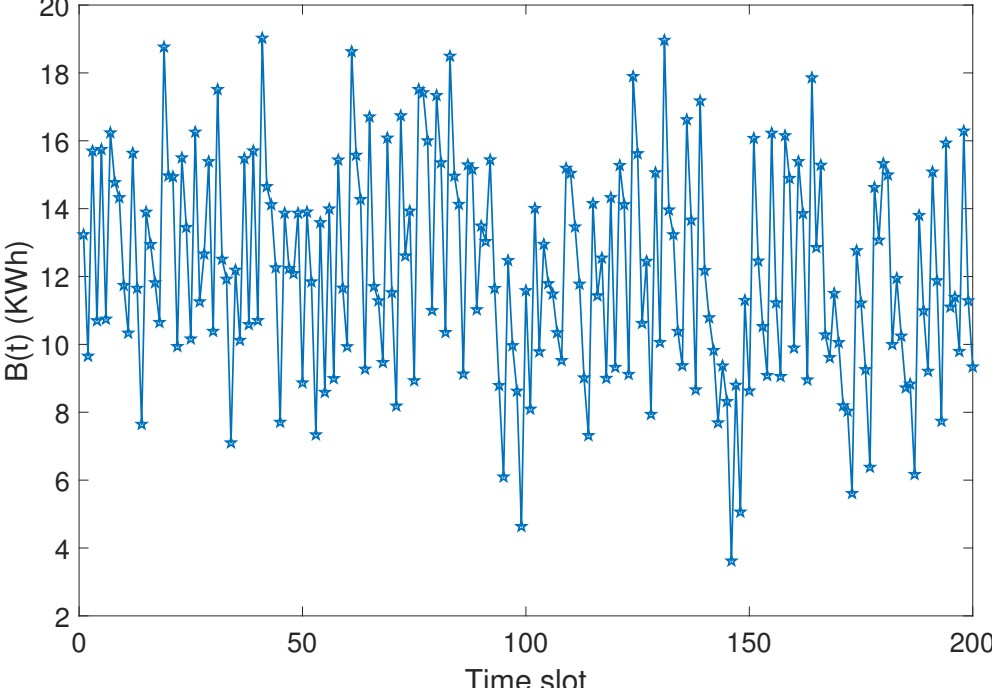

**Figure 5.** The result of battery level of ESS.

We have displayed the percentage of reduced cost in total cost for different deadlines with the parameter $V = 6$ in Figure 6. From examining Figure 6, it is evident that increasing the deadline for EL loads results in a higher percentage of reduced cost in the total cost. Specifically, we can observe that when we set the deadline $d_k^t = 14$, the reduced cost of EL loads accounts for 12.49% of the total cost. Thus, it is apparent that greater benefits can be obtained with longer delays for EL loads. We compare the costs of RL loads and EL loads in each time slot using our algorithm, as shown in Figure 7, for a deadline of $d_k^t = 5$. It is apparent that, under the same conditions, one EL load has a lower cost than one RL load. Figure 7 depicts the percentage of cost savings versus the ratio of EL/(EL + RL) for the deadline $d_k^t = 5, 10, 15$ and the parameter $V = 6$. The percentage of cost savings increases with the increase in the deadline $d_k^t$. We can see that our algorithm will lead to a higher reduced cost in the case of more EL loads. This is because the algorithm is designed to identify and eliminate unnecessary electricity loads. When there are more EL loads, the algorithm will have more opportunities to find and eliminate unnecessary electricity loads. This will lead to a higher reduced cost. Our optimization algorithm works better when there is more elastic load. This is because elastic load can be more flexibly adjusted to meet changing demand. When demand increases, elastic load can be increased to meet the demand. When demand decreases, elastic load can be decreased to avoid wasting resources. Our optimization algorithm can use the flexibility of elastic load to improve the efficiency of the system.

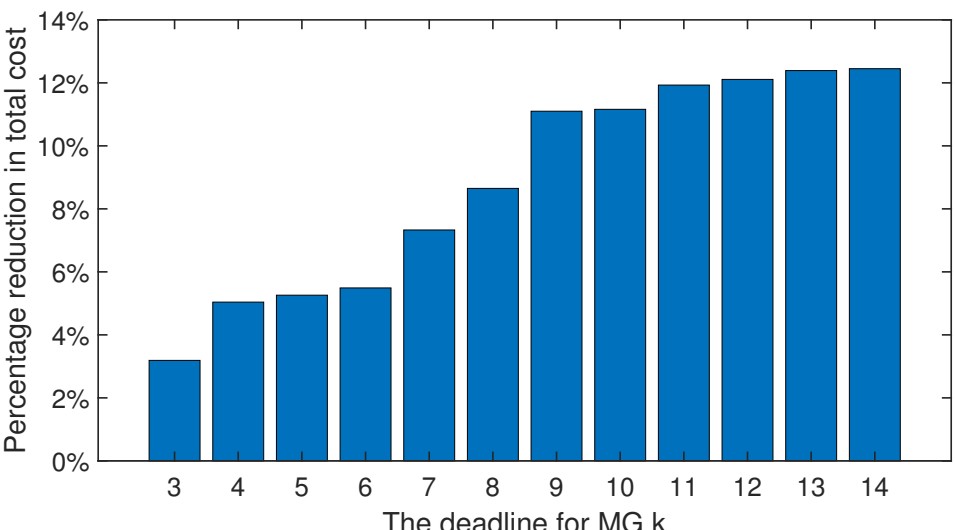

**Figure 6.** The percentage of the reduced cost.

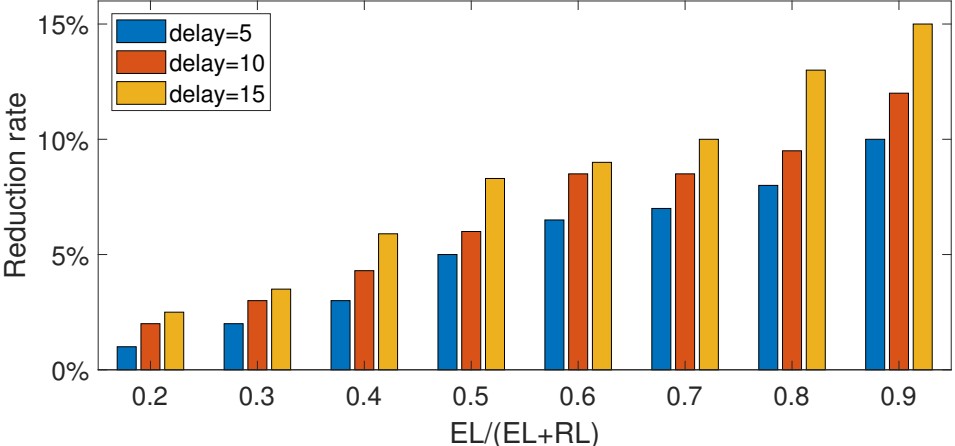

**Figure 7.** Reduction in total cost versus EL/(EL + RL).

Figure 8 depicts the system cost versus the parameter *V* for different load rate EL/(EL + RL) = 0.2, 0.5, 1. The total cost decreases as load rate EL/(EL + RL) increases. When the parameter *V* reaches 30, the total cost decreases slowly as the parameter *V* grows. The system cost also decreases slowly as ESS capacity $b_{max}$ increases, which eventually tends to be a limit. From Figure 5, we can see that the battery level queue $B(t)$ is stable. Our optimization algorithm can allocate the elastic load to those time periods with high demand and remove it from those time periods with low demand. This can help the system use resources more efficiently and lower costs. In Figure 9, we compare our NRA algorithm with the rolling online control (OA) algorithm according to the reference [19]. For each load rate EL/(EL + RL), the total cost by the OA algorithm is higher than that by the NRA algorithm. From Figure 9, we achieve the optimal system cost when *V* = 35. This performance result shows that our proposed NRA can achieve the minimum system cost with the constraint of keeping the battery level queue and ELs' satisfaction queue stable. The simulation result of cumulative reward has been shown in Figure 10. The simulation performance about cumulative profit according to different discount factors $\gamma = 0.005, 0.01, 0.05$ was shown in Figure 10. Cumulative reward plays a very important role in Q-learning. It can help Q-learning algorithms converge to the optimal policy faster. Q-learning algorithms learn the optimal policy by trial and error. In each trial, the Q-learning algorithm calculates a reward for the current state and action. This reward is accumulated into a value called the cumulative reward. The cumulative reward reflects the total reward from one state to another. It can help Q-learning algorithms determine which actions are worth trying. If an action has a high cumulative reward, then the Q-learning algorithm is more likely to try that action. From Figure 11, the total system cost of a community with an energy-sharing algorithm is lower than that of a non-sharing community. Our ESA algorithm can reduce 10% economic cost compared to the NRA Algorithm in 200 time slots. ESA algorithm works by iteratively updating a value function that represents the expected return for taking a particular action in a particular state. The value function is updated based on the reward for taking different actions in different states. The complexity of the Q-learning-based ESA algorithm mainly depends on the following factors: state space, action space, and learning rate. The larger the state space, the greater the amount of computation involved in the Q-learning-based algorithm. The larger the action space, the greater the amount of computation involved in the ESA algorithm. The higher the learning rate, the more computation-intensive the ESA algorithm will be.

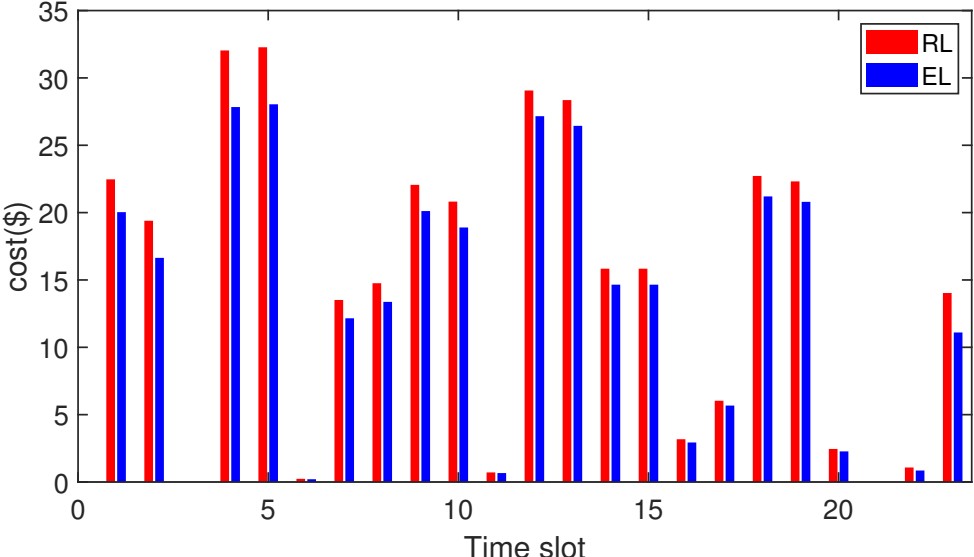

**Figure 8.** System cost of load rate EL/RL.

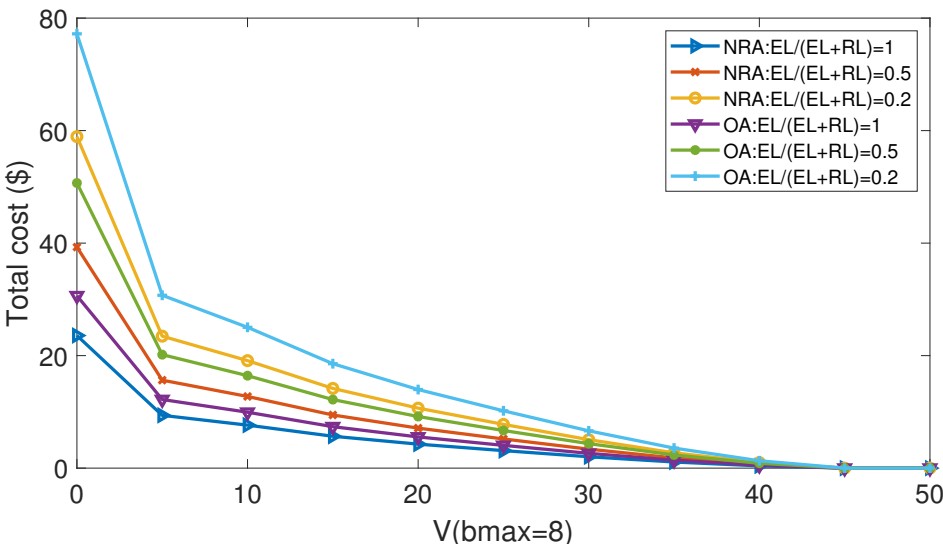

**Figure 9.** System cost versus *V* with different ratio of EL/RL.

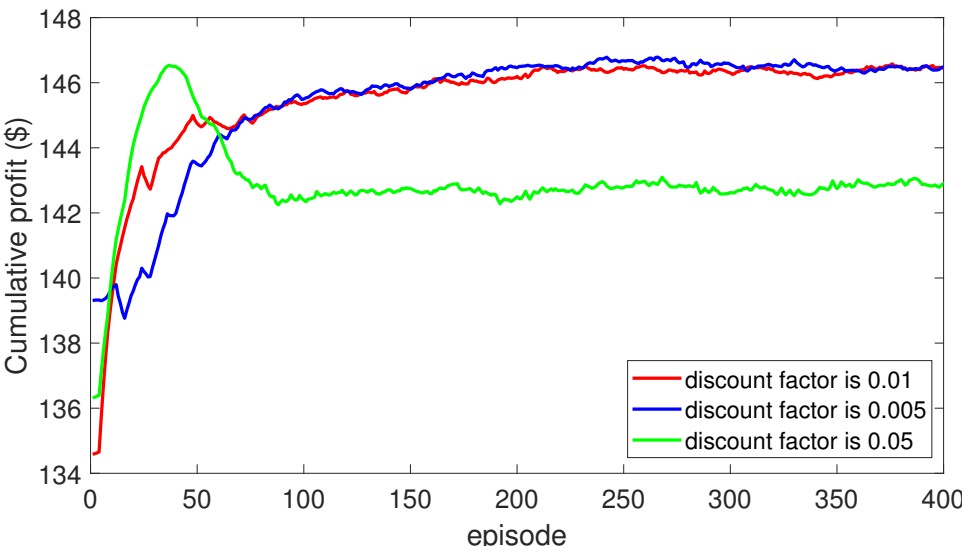

**Figure 10.** Cumulative reward function.

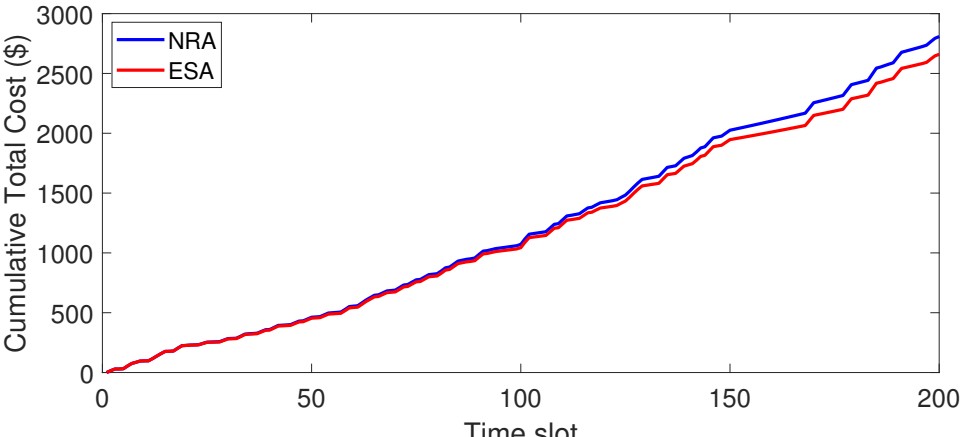

**Figure 11.** System cost under energy-sharing and non-sharing scenarios.

## 5. Conclusions

In this paper, we address the problem of carbon management and resource allocation in an intelligent community with CHP and solar power while taking into account unpredictable power demands and the constraint of carbon emission. We design the non-sharing algorithm by utilizing a Lyapunov optimization approach to solve the stochastic non-convex optimization problem. To facilitate energy sharing, we develop an energy-sharing algorithm under carbon emission constraints based on the Q-learning algorithm. The effectiveness of our proposed energy-sharing algorithm is demonstrated through extensive simulations, which show that it achieves lower costs compared to the non-sharing algorithm. The NRA algorithm shows that a larger ESS maximum output and *V* will lead to larger cost savings. The results show that the ESA algorithm can effectively reduce the cost by 10% of the system, compared with the NRA algorithm. The NRA algorithm satisfies the EL demand before user-defined deadlines and we can see that with the increase in the deadline, the saved cost will increase. The NRA algorithm is a good choice for ESSs to reduce the cost of the system. By the ESA algorithm, energy sharing can help reduce energy costs by allowing producers and consumers to take advantage of economies of solar energy.

**Author Contributions:** Conceptualization, Y.C. and C.Z.; methodology, Y.C.; software, Y.C.; validation, Y.C.; formal analysis, Y.C.; investigation, Y.C.; resources, Y.C.; data curation, Y.C.; original draft preparation, Y.C.; review and editing, Y.C.; visualization, D.L.; supervision, C.Z. and D.L.; project administration, Y.C.; funding acquisition, Y.C. All authors have read and agreed to the published version of the manuscript.

**Funding:** This work was funded by the Fundamental Research Funds for the Central Universities (CUSF-DH-D-2018093); Shanghai Science and Technology Innovation Action Plan Morning Star Project (Sail Special) (22YF1411900); the 4th Batch of Special Grants from China Postdoctoral Science Foundation (2022TQ0210); National Social Science Foundation of China (20&ZD199); National Social Science Fund Major Project of China on "Legalization of Technology Standards for Public Data in China" (21&ZD200); the Key interdisciplinary project for the Central Universities (223201800084); the Humanities and Social Science Research Project of Ministry of Education (20YJC820030).

**Institutional Review Board Statement:** Not applicable.

**Informed Consent Statement:** Not applicable.

**Data Availability Statement:** Not applicable.

**Conflicts of Interest:** The authors declare no conflict of interest.

## Abbreviations

The following abbreviations are used in this manuscript:

| Variables | Description |
|---|---|
| $u^k(t)$ | natgas consumed by the CHP system for a given time slot $t$ |
| $p^k(t)$ | energy sourced from the power grid |
| $b^k(t)$ | battery level |
| $e^k(t)$ | electric power demand |
| $h^k(t)$ | thermal demand |
| $s_k^t$ | delay |
| $d_k^t$ | load's cutoff time |
| $r_s^k(t)$ | solar power derived to the battery |
| $g^k(t)$ | energy dispatched from the boiler to fulfil the heating requirement |
| $\lambda_e(t)$ | electricity price |
| $\lambda_g$ | natgas price |
| $C_k^{Em}$ | peak carbon dioxide emissions from MG $k$ |
| $\omega_p$ | carbon dioxide emissions from electricity generation |
| $\omega_g$ | carbon dioxide emissions from natural gas |
| $B^k(t)$ | energy stored in the ESS during time slot $t$ for MG $k$ |
| $\varsigma_j^k(t)$ | energy sharing between MG $k$ and MG $j$ in the time slot $t$ |

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
