# Peer review of "Carbon Management for Intelligent Community with Combined Heat and Power Systems"

_sustainability, doi:10.3390/su151713257_

Round 1

Reviewer 1 Report

1. In line 71, What does MG stand for?
2. In line 88, what is natgas?
3. Lines from 25 to 31 doesn't give any meaning to this paper. It is mentioned, 'We consider the internal combustion engine CHP system in this paper'. How?
4. The introduction is not clear. It is suggested to write the introduction in a state of art manner to convey the present level of the research domain. Clearly indicate the novelty of your work in line with existing literature. Write about the meaning of 'intelligent community' with suitable reference.
5. Throughout manuscript, number of equations are used. Cite the suitable references for each of them.
6. Section 1 talk about problem formulation? Section 2 heading also convey the same. Check and change them suitably.
7. Add the list of symbols, abbreviations at the end of manuscript.
8. Under section 2.1, two theorems have given and proved. Are they formulated by the authors or universal? If they are universal, no need to prove them here.
9. What was the tool used for numerical simulation? What was the convergence criteria? How was it validated?
10. Compare the results with the previous literature and explain in discussion section.
11. Conclusion looks repetition of the objective. Re-write it with significant findings of the work both quantitative and qualitative manner. 

Language quality is acceptable with minor revision.

Author Response

Reply to Reviewer 1

Comment 1:

In line 71, What does MG stand for?

Response of the authors: Thanks for your valuable comments and suggestions. We have clarified the full name microgrid for the abbreviation MG.

Author action:  

Page 2, Line 71, An optimal scheduling method for a microgrid (MG) with CHP system using model predictive control is proposed in [10] to improve its efficiency and economic performance.

Comment 2:

  1. In line 88, what is natgas?

Response of the authors: Thanks for your valuable comments and suggestions. We have clarified the full name natural gas for the abbreviation natgas.

Author action:  

Page 2, Line 88, including expenses related to power grid and natural gas (natgas) consumption.

Comment 3:

Lines from 25 to 31 doesn't give any meaning to this paper. It is mentioned, 'We consider the internal combustion engine CHP system in this paper'. How?

Response of the authors: Thanks for your valuable comments and suggestions. We have revised Lines from 25 to 31 as “The CHP system mentioned in this paper is the internal combustion engine CHP system”.

Author action: 

Page 1, Line 25, The CHP system mentioned in this paper is the internal combustion engine CHP system.

Comment 4:

The introduction is not clear. It is suggested to write the introduction in a state of art manner to convey the present level of the research domain. Clearly indicate the novelty of your work in line with existing literature. Write about the meaning of 'intelligent community' with suitable reference.

Response of the authors: Thanks for your valuable comments and suggestions. We

have clarified that Intelligent community utilizes emerging technologies such as smart phones, embedded computers, network technology, and radio frequency identification technology to make the entire community management more intelligent and improve the quality of life of residents.

Author action: 

Page 1, Line 18,

Intelligent community utilizes emerging technologies such as smart phones, embedded computers, network technology, and radio frequency identification technology to make the entire community management more intelligent and improve the quality of life of residents.

Page 2, Line 43,

In the following two paragraphs, we briefly discuss the related work about the carbon management for intelligent community with CHP.

Page 2, Line 46,

There are some related works about the energy management of CHP systems to improve the efficiency of power system.

Page 2, Line 56,

These related works didn't consider the heating compensation mechanism between CHP units and heat storages.

Page 2, Line 68,

Some related works have designed algorithms to assist practical systems in making decisions according to energy storage, utilization or selling to the grid based on real-time prices (spot prices).

Page 2, Line 75,

The next four papers assumed predictability of future electricity prices and load demand, and focused solely on proposing optimal task scheduling algorithms according to consumer convenience.

Page 2, Line 87,

However, these papers did not consider carbon management with the heterogeneous load demands. This paper aims to study the issue of carbon management and resource allocation with CHP for heterogeneous load demands in an intelligent community to minimize the system costs associated with power grid and gas.

Comment 5:

Throughout manuscript, number of equations are used. Cite the suitable references for each of them.

Response of the authors: Thanks for your valuable comments and suggestions.

We have added the reference for Eqn. (4) in the revised manuscript.

Comment 6:

Section 1 talk about problem formulation? Section 2 heading also convey the same. Check and change them suitably.

Response of the authors: Thanks for your valuable comments and suggestions. We have merged Section 2 into Section 1.

Comment 7:

  1. Add the list of symbols, abbreviations at the end of manuscript.

Response of the authors: Thanks for your valuable comments and suggestions. We have added the abbreviations in Table 1 at the end of manuscript.

Comment 8:

Under section 2.1, two theorems have given and proved. Are they formulated by the authors or universal? If they are universal, no need to prove them here.

Response of the authors: Thanks for your valuable comments and suggestions. The theorems are formulated by the authors.

Comment 9:

What was the tool used for numerical simulation? What was the convergence criteria? How was it validated?

Response of the authors: Thanks for your valuable comments and suggestions. We have add the tool Matlab 2021 on Intel Core i7 used for numerical simulation. We have added the description of the convergence of our method. Lyapunov optimization method ensures the stability and convergence of the system by designing Lyapunov function. In the stable region, the value of Lyapunov function is positive; In the unstable region, the value of the Lyapunov function is negative. By designing the Lyapunov function, the system can be guaranteed to be in a stable state in the stable region.

Author action: 

Page 11, Line 255,

In this section, we will evaluate the performance of the NRA algorithm and the ESA algorithm with real-time electricity price by Matlab 2021 on Intel Core i7.

Page 10, Line 206,

Lyapunov optimization method ensures the stability and convergence of the system by designing Lyapunov function. In the stable region, the value of Lyapunov function is positive; In the unstable region, the value of the Lyapunov function is negative. By designing the Lyapunov function, the system can be guaranteed to be in a stable state in the stable region.

Comment 10:

Compare the results with the previous literature and explain in discussion section.

Response of the authors: Thanks for your valuable comments and suggestions.

We have added the performance comparison of simulation results from previous literature. In Fig. 8, we compare our NRA algorithm with rolling online control (OA) algorithm according to the reference [17]. For each load rate EL/(EL+RL), the total cost by OA algorithm is higher than that by NRA algorithm.

Author action: 

Page 16, Line 326, In Fig. 8, we compare our NRA algorithm with rolling online control (OA) algorithm according to the reference [17]. For each load rate EL/(EL+RL), the total cost by OA algorithm is higher than that by NRA algorithm.

Figure 8. Cost versus V with different ratio of EL/RL.

Comment 11:

  1. Conclusion looks repetition of the objective. Re-write it with significant findings of the work both quantitative and qualitative manner.

Response of the authors: Thanks for your valuable comments and suggestions.

We have rewrite the conclusion in quantitative and qualitative manner.

Author action: 

Page 16, Line 344,

In this paper, we address the problem of carbon management and resource allocation in an intelligent community with CHP and solar power, while taking into account unpredictable power demands and the constraint of carbon emission. We design the non-sharing algorithm by utilizing a Lyapunov optimization approach to solve the stochastic non-convex optimization problem. To facilitate the energy sharing, we develop an energy-sharing algorithm under carbon emission constraints based on the Q-learning Algorithm. The effectiveness of our proposed energy-sharing algorithm is demonstrated through extensive simulations, which show that it achieves lower costs compared to the non-sharing algorithm. The NRA algorithm shows that a larger ESS maximum output and V will lead to a larger cost savings. The results show that the ESA algorithm can effectively reduce the cost 10% of the system, compared with NRA algorithm. The NRA algorithm satisfies the EL demand before user-defined deadlines and we can see that with the increase of the deadline, the saved cost will increase. The NRA algorithm is a good choice for ESSs to reduce the cost of the system. By the ESA algorithm, energy sharing can help reduce energy costs by allowing producers and consumers to take advantage of economies of solar energy.

Reviewer 2 Report

The paper addresses carbon management and resource allocation in intelligent communities with combined heat and power systems and solar power, aiming to minimize system costs over time. It uses Lyapunov optimization for carbon management and proposes a Q-learning-based energy sharing algorithm.

The paper exhibits good writing, organization, and structural clarity. The problem formulation and system model are well-elaborated. However, some improvements are needed:

1)   Ensure that abbreviations are introduced upon their first use (e.g., MG, NRA, PG&E).

2)   Efforts should be made to reduce redundancy ( For example the repetition of "Fig. 6" within the span of Line 299 to 303) Consider rephrasing to enhance clarity and conciseness.

3)   Pay attention to the notation and related description of figures, specifically in the numerical simulation section (e.g., Fig.7).

4)   Discussion and justification of the use of a Q-learning-based algorithm that adds complexity to the problem.

5)   To better understand the case study (Numerical Simulation section), the author can provide a clearer description of the characteristics of the studied system (production sources, load profiles….).

6)   While the paper's core focus centers on carbon management, it's worth noting that the simulation lacks the necessary quantification of carbon-related parameters, potentially limiting the depth of insights provided.

Author Response

Reply to Reviewer 2

General Comment:

The paper addresses carbon management and resource allocation in intelligent communities with combined heat and power systems and solar power, aiming to minimize system costs over time. It uses Lyapunov optimization for carbon management and proposes a Q-learning-based energy sharing algorithm.

The paper exhibits good writing, organization, and structural clarity. The problem formulation and system model are well-elaborated.

Author response: 

Thank you so much for your pertinent summary of our main results. Your recognition of our contributions is particularly appreciated.

Comment 1:

Ensure that abbreviations are introduced upon their first use (e.g., MG, NRA, PG&E).

Response of the authors: Thanks for your valuable comments and suggestions. We have clarified the full name microgrid for the abbreviation MG, the full name Pacific Gas and Electric Company for the abbreviation PG&E, the full name non-sharing resource allocation algorithm for the abbreviation NRA.

Author action:  

Page 2, Line 71, An optimal scheduling method for a microgrid (MG) with CHP system using model predictive control is proposed in [10] to improve its efficiency and economic performance.

Page 8, Line 180, non-sharing resource allocation algorithm (NRA)

Page 12, Line 264, Pacific Gas and Electric Company (PG&E)

Comment 2:

Efforts should be made to reduce redundancy ( For example the repetition of "Fig. 6" within the span of Line 299 to 303) Consider rephrasing to enhance clarity and conciseness.

Response of the authors: Thanks for your valuable comments and suggestions. We have reduced the redundancy of Figure. 6 to enhance clarity and conciseness.

Author action:  

Page 15, Line 308,

It is apparent that, under the same conditions, one EL load have a lower cost than one RL load. Fig. 6 depicts the percentage of cost savings versus the ratio of EL/(EL+RL) for the deadline dk=5, 10, 15 and the parameter V=6. The percentage of cost savings increases with the increase of the deadline dk.

Comment 3:

Pay attention to the notation and related description of figures, specifically in the numerical simulation section (e.g., Fig.7).

Response of the authors: Thanks for your valuable comments and suggestions. We have added the abbreviations in Table 2 at the end of manuscript. We have added the related description of Figure. 7.

Author action:  

Page 14, Line 284,

Figure 7. System cost of load rate EL/RL.

Page 14, Line 320,

Fig. 7 depicts the system cost versus the parameter V for different load rate EL/(EL+RL)= 0.2, 0.5, 1. The total cost decreases as load rate EL/(EL+RL) increases. When the parameter V reaches 30, the total cost decreases slowly as the parameter V grows.

Comment 4:

Discussion and justification of the use of a Q-learning-based algorithm that adds complexity to the problem.

Response of the authors: Thanks for your valuable comments and suggestions. We have added the discussion of the use of a Q-learning-based algorithm. 

Author action:  

Page 16, Line 343,

ESA algorithm works by iteratively updating a value function that represents the expected return for taking a particular action in a particular state. The value function is updated based on the reward for taking different actions in different states. The complexity of Q-learning-based ESA algorithm mainly depends on the following factors: state space, action space and learning rate. The larger the state space, the greater the amount of computation involved in Q-learning-based algorithm. The larger the action space, the greater the amount of computation involved in ESA algorithm. The higher the learning rate, the more computation-intensive ESA algorithm will be.

Comment 5:

To better understand the case study (Numerical Simulation section), the author can provide a clearer description of the characteristics of the studied system (production sources, load profiles….).

Response of the authors: Thanks for your valuable comments and suggestions. We have added a clearer description of the characteristics. We have added the parameter setting in the numerical simulation section.

Author action:  

Page 13, Line 271,

The carbon dioxide emissions from electricity generation and that from natural gas are denoted as =450g CO2/kWh and =1.885kg CO2/m3.

Comment 6:

While the paper's core focus centers on carbon management, it's worth noting that the simulation lacks the necessary quantification of carbon-related parameters, potentially limiting the depth of insights provided.

Response of the authors: Thanks for your valuable comments and suggestions. We have added the carbon-related parameter setting.

Author action:  

Page 13, Line 271,

The carbon dioxide emissions from electricity generation and that from natural gas are denoted as =450g CO2/kWh and =1.885kg CO2/m3.

Reviewer 3 Report

In this paper, the authors have addressed the problem of carbon management and resource allocation in an intelligent community with CHP and solar power. The overall study is good but it requires a major revision.

1.      First, define all the acronyms mentioned in the text and then use a short form. For example, microgrid (MG) in “An optimal scheduling method for a MG with CHP system using model”

2.      Many language errors throughout the article; rewrite the abstract and conclusion using past tense where applicable. In technical writing, the pronoun "we" is not preferred.

3.      Use the same caption in the Figures as well as in the text (Figure 1 or Fig. 1) throughout the article.

4.      Units in some figures are not appropriately labelled. For example, Figure 9. Cumulative profit in $ ?

5.      The formatting of this manuscript is also poor. Almost all figures are inserted in the middle of the text. First of all, complete the sentence, then insert the figure.  

6.      Some variables are not defined properly in equations.

7.      In introduction draw Q-learning principle diagram and cite some latest and relevant articles related to Q-table. No discussion about Q-values?

8.      A sensitivity analysis should be conducted to identify the impact of the different variables on the systems.

9.      Conducting economic studies in detail is also recommended, along with carbon management.

10.  Flowcharts of the designed algorithms are also required.

11.  The detailed parameters of MGs components in table or diagram are also required.

12.  Axis units are not defined in some figures—for example, Figure 9. Cumulative profit has no units? $?

13.  The results in Figure. 8 show ESA is better than NRA but it contradicts in conclusion “The results show that the NRA algorithm can effectively reduce the cost of the system. The NRA algorithm satisfies the EL demand before user-defined deadlines and we can see that with the increase of the deadline, the saved cost will increase”. Explain.

Many language errors throughout the article; rewrite the abstract and conclusion using past tense where applicable. In technical writing, the pronoun "we" is not preferred.

Author Response

Reply to Reviewer 3

General Comment:

In this paper, the authors have addressed the problem of carbon management and resource allocation in an intelligent community with CHP and solar power. The overall study is good but it requires a major revision.

Author response: Thank you so much for your pertinent summary of our main results. Your recognition of our contributions is particularly appreciated.

Comment 1:

First, define all the acronyms mentioned in the text and then use a short form. For example, microgrid (MG) in “An optimal scheduling method for a MG with CHP system using model”

Response of the authors: Thanks for your valuable comments and suggestions. We have clarified the full name microgrid for the abbreviation MG.

Author action:  

Page 2, Line 71, An optimal scheduling method for a microgrid (MG) with CHP system using model predictive control is proposed in [10] to improve its efficiency and economic performance.

Comment 2:

Many language errors throughout the article; rewrite the abstract and conclusion using past tense where applicable. In technical writing, the pronoun "we" is not preferred.

Response of the authors: Thanks for your valuable comments and suggestions. We have clarified the full name microgrid for the abbreviation MG. We have revised the abstract and conclusion using past tense. We have revise the pronoun “we”.

Author action:  

Page 1, Line 1,

In recent years, solar power technology and energy storage technology have advanced, leading to the increased use of solar power devices and energy storage systems in residential areas. Carbon management has become an important method to help the community manager timely guide energy consumption, effectively reduce the carbon emissions of the community, and reduce the substantial harm to the environment. This paper aims to study the issue of carbon management and resource allocation in an intelligent community with combined heat and power (CHP) systems and solar power. The presence of heterogeneous load demands in the power grid was considered. The main objective was to minimize the average system cost over time, which included the costs associated with power grid and gas. The Lyapunov optimization theory was employed to solve the non-convex optimization problem of carbon management and resource allocation without energy sharing. For solving the energy sharing problem, we designed an energy sharing algorithm based on the Q-learning algorithm. Lastly, we conducted extensive simulations using actual trace data to validate the effectiveness of our proposed algorithms.

Page 17, Line 353,

In this paper, we address the problem of carbon management and resource allocation in an intelligent community with CHP and solar power, while taking into account unpredictable power demands and the constraint of carbon emission. We design the non-sharing algorithm by utilizing a Lyapunov optimization approach to solve the stochastic non-convex optimization problem. To facilitate the energy sharing, we develop an energy-sharing algorithm under carbon emission constraints based on the Q-learning Algorithm. The effectiveness of our proposed energy-sharing algorithm is demonstrated through extensive simulations, which show that it achieves lower costs compared to the non-sharing algorithm. The NRA algorithm shows that a larger ESS maximum output and V will lead to a larger cost savings. The results show that the ESA algorithm can effectively reduce the cost 10% of the system, compared with NRA algorithm. The NRA algorithm satisfies the EL demand before user-defined deadlines and we can see that with the increase of the deadline, the saved cost will increase. The NRA algorithm is a good choice for ESSs to reduce the cost of the system. By the ESA algorithm, energy sharing can help reduce energy costs by allowing producers and consumers to take advantage of economies of solar energy.

Page 3, Line 91,

Three contributions of the paper are summarized as follows.

Page 3, Line 103,

Simulations have been conducted by using actual trace data to verify the effectiveness of our proposed algorithms. The energy sharing algorithm was compared with a non-sharing resource allocation algorithm

Page 3, Line 108,

A mathematical model for a power grid that incorporates CHP systems, solar panels, ESS, and boilers was presented in Section II. The solar power sharing strategy and control objectives was introduced.

Comment 3:

Use the same caption in the Figures as well as in the text (Figure 1 or Fig. 1) throughout the article.

Response of the authors: Thanks for your valuable comments and suggestions. We have revised Fig. As Figure to use the same caption.

Comment 4:

Units in some figures are not appropriately labelled. For example, Figure 9. Cumulative profit in $ ?

Response of the authors: Thanks for your valuable comments and suggestions. We have added the units $ in Figure 9.

Comment 5:

The formatting of this manuscript is also poor. Almost all figures are inserted in the middle of the text. First of all, complete the sentence, then insert the figure.

Response of the authors: Thanks for your valuable comments and suggestions. We have adjusted the location of the figures.

Comment 6:

Some variables are not defined properly in equations.

Response of the authors: Thanks for your valuable comments and suggestions. We have clarified We have added the abbreviations in Table 2 at the end of manuscript.

Comment 7:

In introduction draw Q-learning principle diagram and cite some latest and relevant articles related to Q-table. No discussion about Q-values?

Response of the authors: Thanks for your valuable comments and suggestions. We have added the related work about Q-learning algorithm. We have added the discussion of the use of a Q-learning-based algorithm. 

Author action:  

Page 2, Line 71,

Some literature uses reinforcement learning methods to solve energy management problems.  A total cost of ownership model was established in [15]including energy consumption and power source degradation, where Q-learning algorithm is proposed to determine the optimal energy management strategy. A real-time energy management strategy was proposed in [16] by combining Q-Learning method with model predictive control method.

[15] Q. Li, X. Meng, F. Gao, G. Zhang and W. Chen, "Approximate Cost-Optimal Energy Management of Hydrogen Electric Multiple Unit Trains Using Double Q-Learning Algorithm," IEEE Transactions on Industrial Electronics, vol. 69, no. 9, pp. 9099-9110, Sept. 2022.

[16] S. Shen et al., "Real-Time Energy Management for Plug-in Hybrid Electric Vehicles via Incorporating Double-Delay Q-Learning and Model Prediction Control," IEEE Access, vol. 10, pp. 131076-131089, 2022.

Page 16, Line 343,

ESA algorithm works by iteratively updating a value function that represents the expected return for taking a particular action in a particular state. The value function is updated based on the reward for taking different actions in different states. The complexity of Q-learning-based ESA algorithm mainly depends on the following factors: state space, action space and learning rate. The larger the state space, the greater the amount of computation involved in Q-learning-based algorithm. The larger the action space, the greater the amount of computation involved in ESA algorithm. The higher the learning rate, the more computation-intensive ESA algorithm will be.

Comment 8:

A sensitivity analysis should be conducted to identify the impact of the different variables on the systems.

Response of the authors: Thanks for your valuable comments and suggestions. We have added the simulation performance about cumulative profit according to different discount factors =0.005,0.01,0.05 in Figure. 9.

Comment 9:

Conducting economic studies in detail is also recommended, along with carbon management.

Response of the authors: Thanks for your valuable comments and suggestions. We have added the performance comparison of simulation results from previous literature. In Fig. 8, we compare our NRA algorithm with rolling online control (OA) algorithm according to the reference [17]. For each load rate EL/(EL+RL), the total cost by OA algorithm is higher than that by NRA algorithm. We have added the carbon-related parameter setting.

Author action: 

Page 16, Line 326, In Fig. 8, we compare our NRA algorithm with rolling online control (OA) algorithm according to the reference [17]. For each load rate EL/(EL+RL), the total cost by OA algorithm is higher than that by NRA algorithm.

Figure 8. Cost versus V with different ratio of EL/RL.

Page 13, Line 271,

The carbon dioxide emissions from electricity generation and that from natural gas are denoted as =450g CO2/kWh and =1.885kg CO2/m3.

Comment 10:

Flowcharts of the designed algorithms are also required.

Response of the authors: Thanks for your valuable comments and suggestions. We have added the flowcharts of NSA algorithm in Figure 2.

Figure 2. Flow chart of energy sharing algorithm.

Comment 11:

The detailed parameters of MGs components in table or diagram are also required.

Response of the authors: Thanks for your valuable comments and suggestions. We have added a clearer description of the characteristics. We have added the parameter setting in the numerical simulation section.

Comment 12:

Axis units are not defined in some figures—for example, Figure 9. Cumulative profit has no units? $?

Response of the authors: Thanks for your valuable comments and suggestions.We have added the units $ in Figure 9.

Comment 13:

The results in Figure. 8 show ESA is better than NRA but it contradicts in conclusion “The results show that the NRA algorithm can effectively reduce the cost of the system. The NRA algorithm satisfies the EL demand before user-defined deadlines and we can see that with the increase of the deadline, the saved cost will increase”. Explain.

Response of the authors: Thanks for your valuable comments and suggestions. We have added the performance comparison of simulation results from previous literature. In Fig. 8, we compare our NRA algorithm with rolling online control (OA) algorithm according to the reference [17]. For each load rate EL/(EL+RL), the total cost by OA algorithm is higher than that by NRA algorithm. We also rewrite the conclusion.

Author action: 

Page 16, Line 326, In Fig. 8, we compare our NRA algorithm with rolling online control (OA) algorithm according to the reference [17]. For each load rate EL/(EL+RL), the total cost by OA algorithm is higher than that by NRA algorithm.

Figure 8. Cost versus V with different ratio of EL/RL.

Page 17, Line 353,

In this paper, we address the problem of carbon management and resource allocation in an intelligent community with CHP and solar power, while taking into account unpredictable power demands and the constraint of carbon emission. We design the non-sharing algorithm by utilizing a Lyapunov optimization approach to solve the stochastic non-convex optimization problem. To facilitate the energy sharing, we develop an energy-sharing algorithm under carbon emission constraints based on the Q-learning Algorithm. The effectiveness of our proposed energy-sharing algorithm is demonstrated through extensive simulations, which show that it achieves lower costs compared to the non-sharing algorithm. The NRA algorithm shows that a larger ESS maximum output and V will lead to a larger cost savings. The results show that the ESA algorithm can effectively reduce the cost 10% of the system, compared with NRA algorithm. The NRA algorithm satisfies the EL demand before user-defined deadlines and we can see that with the increase of the deadline, the saved cost will increase. The NRA algorithm is a good choice for ESSs to reduce the cost of the system. By the ESA algorithm, energy sharing can help reduce energy costs by allowing producers and consumers to take advantage of economies of solar energy.

Round 2

Reviewer 1 Report

The paper can be accepted in the present form.

Reviewer 3 Report

The authors addressed all my comments and the overall quality of the manuscript has been improved. The paper may be published in its current form or the paper may also be published after proper adjusting the location of Figures  3 to 10 after the text.

Quality of English is improved.